# The characteristics and extent of food industry involvement in peer-reviewed research articles from 10 leading nutrition-related journals in 2018

**Gary Sacks**[1]*, **Devorah Riesenberg**[1], **Melissa Mialon**[2], **Sarah Dean**[1], **Adrian J. Cameron**[1]

**1** Global Obesity Centre (GLOBE), Institute for Health Transformation, School of Health and Social Development, Faculty of Health, Deakin University, Geelong, Victoria, Australia, **2** School of Public Health, University of Sao Paulo, Sao Paulo, Brazil

* gary.sacks@deakin.edu.au

**Data Availability Statement:** All relevant data are within the paper and its Supporting Information files.

## Abstract

### Introduction

There is emerging evidence that food industry involvement in nutrition research may bias research findings and/or research agendas. However, the extent of food industry involvement in nutrition research has not been systematically explored. This study aimed to identify the extent of food industry involvement in peer-reviewed articles from a sample of leading nutrition-related journals, and to examine the extent to which findings from research involving the food industry support industry interests.

### Methods

All original research articles published in 2018 in the top 10 most-cited nutrition- and dietetics-related journals were analysed. We evaluated the proportion of articles that disclosed involvement from the food industry, including through author affiliations, funding sources, declarations of interest or other acknowledgments. Principal research findings from articles with food industry involvement, and a random sample of articles without food industry involvement, were categorised according to the extent to which they supported relevant food industry interests.

### Results

196/1,461 (13.4%) articles reported food industry involvement. The extent of food industry involvement varied by journal, with *The Journal of Nutrition* (28.3%) having the highest and *Paediatric Obesity* (3.8%) having the lowest proportion of industry involvement. Processed food manufacturers were involved in the most articles (77/196, 39.3%). Of articles with food industry involvement, 55.6% reported findings favourable to relevant food industry interests, compared to 9.7% of articles without food industry involvement.

**Funding:** GS and AJC were supported by Heart Foundation Future Leader Fellowships (102035 and 36357, respectively) from the National Heart Foundation of Australia (https://www.heartfoundation.org.au/). GS and AJC are researchers within a National Health and Medical Research Council (NHMRC) (https://www.nhmrc.gov.au/) Centre of Research Excellence in Food Retail Environments for Health (RE-FRESH) (APP1152968) (Australia). GS is also a researcher within a NHMRC Centre for Research Excellence entitled Reducing Salt Intake Using Food Policy Interventions (APP1117300). The authors are solely responsible for the opinions, hypotheses and conclusions or recommendations expressed in this publication, and they do not necessarily reflect their funders' vision. The funders had no role in study design, data collection and analysis, decision to publish, or preparation of the manuscript.

**Competing interests:** GS and AJC are academic partners on a publicly funded healthy supermarket intervention trial that includes Australian local government and supermarket retail (IGA) collaborators. GS has been involved in studies to benchmark the policies and commitments of food companies related to obesity prevention and nutrition in Australia, New Zealand, Canada, Malaysia and Europe. The authors have not received funding from any organization in the food industry. The authors have no other potential competing interests to declare. The competing interests of the authors do not alter our adherence to PLOS ONE policies on sharing data and materials.

## Conclusion

Food industry involvement in peer-reviewed research in leading nutrition-related journals is commonplace. In line with previous literature, this study has shown that a greater proportion of peer-reviewed studies involving the food industry have results that favour relevant food industry interests than peer-reviewed studies without food industry involvement. Given the potential competing interests of the food industry, it is important to explore mechanisms that can safeguard the integrity and public relevance of nutrition research.

## Introduction

Dietary risk factors are associated with more deaths and disability worldwide than any other modifiable factor [1]. A key driver of poor diets globally has been a nutrition transition characterised by increased consumption of ultra-processed packaged foods [2–4]. These foods are manufactured, marketed and sold by a diverse selection of companies and organisations, collectively referred to as the 'food industry' [5]. Importantly, global food systems are now dominated by a relatively small number of large transnational food companies [2, 6]. The continued generation of profit by these large food companies typically relies on aggressive marketing of their products and brands, as well as political strategies to create regulatory environments that facilitate their market power [7].

Nutrition research is fundamental to efforts to promote healthy eating behaviours and health. However, there is concern regarding how the involvement of the food industry in nutrition research affects the nature of studies conducted, the nutrition research agenda and the findings of individual studies [8–10]. The interests of many commercial food industry actors are misaligned with clinical and public health objectives as the legal mandate of corporations is to return profit for their shareholders, without explicit consideration of broader social impact [10, 11]. In recognition of the inherent risks and to preserve the scientific credibility of nutrition-related research, food industry involvement in research is increasingly scrutinized [12, 13].

Food industry involvement in research can take many forms. These forms of involvement include, amongst others, the provision of funding and the involvement of food company employees as part of research teams. There are many reasons why food companies might be involved in nutrition-related research. These reasons may include unobjectionable motives such as a willingness to develop new knowledge, assist in research translation and contribute expertise and resources [14]. However, from a public health perspective, several concerns have been identified regarding food industry involvement in research. These include: 1) the creation of increased marketing opportunities for industry products, many of which are harmful to population health [15]; 2) the establishment and nurturing of relationships between the food industry and nutrition researchers that serves to increase perceived industry credibility, reduce industry criticism, and encourage increased dependency on the food industry [16, 17]; 3) industry influence over research agendas to preferentially focus on topics likely to benefit industry interests, rather than topics of public health importance [18]; 4) industry influence on the methods, conclusions and impact of research in ways that are likely to favour industry interests over and above other factors [9, 19–21]; and 5) use of research for political purposes [22, 23]. An increased dependence on food industry funding by academics has been documented [9, 12, 16, 24], with food industry funding sometimes acknowledged as a strategically important funding source for the university sector [25].

Previous research has investigated the impact of food industry sponsorship on the findings of published research. Several studies have found that papers sponsored by the food industry typically favour industry interests [9, 21, 26], although a recent meta-analysis found that the quantitative difference in conclusions between food industry-sponsored and non–industry-sponsored nutrition studies was not significant [8]. To date, no study has comprehensively examined the extent and nature of food industry involvement in peer-reviewed research. Better information regarding the extent of food industry involvement, characteristics (e.g., industry sector, company size) of food industry actors that are involved in nutrition-related research, and the ways in which they are involved (e.g., study authorship, different types of funding provided) would assist efforts to assess and manage the potential impact and implications of food industry involvement in research.

This study aimed to contribute to a growing body of empirical evidence related to food industry involvement in peer-reviewed published research by systematically identifying the extent of food industry involvement in research articles from a large sample of leading nutrition-related journals. In addition, this study examined the extent to which research findings support food industry interests for both articles with declared food industry involvement, and those with no declared food industry involvement.

## Methods

### Sample

The study examined articles published in 2018 in the top 10 nutrition and dietetics journals as defined by the SCImago Journal ranking (SJR) as at June 2019. The SJR is a measure of a journal's impact, and expresses the average number of weighted citations received in a selected year by the documents published in the journal in the three previous years [27]. The selected journals included (in alphabetical order): *Advances in Nutrition*, *Clinical Nutrition*, *International Journal of Behavioural Nutrition and Physical Activity*, *International Journal of Obesity*, *Nutrition Research Reviews*, *Nutrition Reviews*, *Obesity*, *Paediatric Obesity*, *The American Journal of Clinical Nutrition* and *The Journal of Nutrition*.

Details of all articles (n = 1,732) published in the selected journals in 2018 were extracted from Medline, CINAHL, Global Health or PubMed. Article types included in the study were original research articles, reviews, short/brief reports and short communications. Article types excluded were errata/corrections, editorials, perspectives, letters to the editor and other related article types. We also examined the disclosed conflicts of interest of the editorial board of each of the selected journals (based on information provided on the website of each journal), links of the selected journals and their editors to the food industry (based on biographical information provided on the journal website and/or on the website of each editor's primary affiliation), as well as each journal's requirements for authors to disclosure conflicts of interest and any other related policies (based on information provided on the website of each journal).

### Food industry involvement

Each included article was examined independently by two of the authors (DR and GS) to determine whether there was food industry involvement in the paper. For the purposes of this study, the "food industry" was broadly defined to include all organisations involved in food and non-alcoholic beverage production, distribution, marketing and retail, as well as relevant industry groups and trade associations [28]. We included manufacturers of dietary supplements and breast-milk substitutes in this definition. In recognition of the known industry tactic of establishing 'front groups' (defined as an organisation that purports to represent one agenda while in reality it serves some other party or interest whose sponsorship is hidden or

rarely mentioned) [29], our definition of "food industry" also included organisations that received the majority of their funding from the food industry.

Food industry involvement was determined based on examination of author affiliations, declared funding sources, declarations of interests, and acknowledgements within each article. All organisations identified through these sections of each article were assessed to determine whether they could be classified as part of the food industry. All universities were considered as not part of the food industry. Organisations known by the authors to be part of the food industry as well as those on an established list of known food industry front groups were classified as such [30]. Searches of the primary websites of all other organisations were conducted to determine the nature of their operations and their funding sources, where relevant, in order to determine if they could be considered as part of the food industry [31].

Food industry actors identified through the study were classified into one of nine different sectors: 1) dairy; 2) dietary supplement manufacturing; 3) food chemical suppliers and food technology companies; 4) food retail; 5) meat and livestock; 6) non-alcoholic beverage manufacturing; 7) primary production (non-dairy, non-meat); 8) processed food manufacturing; and 9) other food industry organisations (see **S1 Table** for definitions of what was included in each sector). Categorisations were based on an assessment of the primary areas of activity of the actor, based on the knowledge of the authors and information provided on the website of the actor. In addition, we classified food industry actors into three categories based on the size and nature of their operations. These included large corporations (with annal global revenue > USD1 billion), trade/industry associations, and small corporations/other entities (annual global revenue < USD1 billion). This classification was based on information obtained from the Euromonitor *Passport* database [32], supplemented by internet searches of the name of the food industry actor where necessary. All categorisation of food industry actors was performed independently by two of the authors (DR and SD), with any discrepancies discussed and resolved with a third author (GS).

Based on the information extracted, papers were categorised as having food industry involvement if: 1) any of the authors self-affiliated as an employee, member or representative of the food industry; 2) the authors declared funding from the food industry, including direct funding for the study, donation of products to be used for the study, or funding received for other activities (e.g., conference attendance) not directly related to the study; or 3) other stated food industry involvement (e.g., through conflicts noted in the acknowledgments sections or other involvement that did not fit within the other categories). Where an individual article included multiple forms of industry involvement, each form of involvement was noted.

## Classification of principal findings

The 'principal findings' of all articles that had involvement with the food industry were classified according to whether the findings were: 1) favourable to the interests of the food industry actor; 2) unfavourable to the interests of the food industry actor; 3) mixed; 4) neutral; or 5) not applicable to the food industry actor/s involved (see Table 1 for definition of each classification). The principal findings were operationalised as the results that were reported in the 'results' section of the abstract of the paper. If the relevant section of the abstract contained insufficient information to deduce the nature of the principal findings, the 'results' and 'discussion' sections of the paper were also examined to understand the nature of the principal reported findings. This approach was based on methods previously used for similar types of analyses [8, 33].

For each of the ten journals, a sample of randomly selected original research articles that did not report food industry involvement was also selected. The process for selection of these

**Table 1. Definitions used to classify principal findings of articles examined.**

| Classification of principal findings | Definition | Examples of how this was operationalised |
| --- | --- | --- |
| Favourable to the interests of the food industry actor | The principal findings were favourable to a specific product (or group of products) relevant to the actor/s and/or they were favourable to the interests of the food industry actor/s more generally. | The authors concluded that the product had beneficial health effects or cast doubts on the evidence linking the product to health harms e.g. for papers that reported involvement of the dairy industry, the principal findings focused on either the benefits of the consumption of dairy products or calcium on cardiovascular health. |
| Unfavourable to the interests of the food industry actor | The principal findings did not support the food industry actor/s and/or consumption of relevant products. | The authors concluded that the product did not bring beneficial health effects e.g. for papers that reported involvement of the non-alcoholic beverage industry, the principal findings focused on either the harmful effects of sugar consumption or non-nutritive sweeteners. |
| Mixed | The principal findings included both favourable and unfavourable results with respect to the interests of the food industry actor/s. | The authors discussed both positive and negative health effects of the product e.g. papers that found that consuming a product (e.g. red meat or refined carbohydrates) has both positive and negatives effects on the consumer. |
| Neutral | The principal findings were neither favourable nor unfavourable to the food industry actor/s. | Descriptive findings or trends in consumption of a particular product. |
| Not applicable to the food industry actor/s involved | The research question and related findings had no apparent relevance to the involved industry actor/s. | The paper did not focus on a product or a component of a product relevant to the food industry. |

articles was that, first, the number of articles with food industry involvement for each journal was calculated. Then, the matching number of articles from each journal, but without food industry involvement, was selected randomly from the list of included articles using the RAND function in Excel. Accordingly, an equal number of articles with and without industry involvement in each journal was selected for analysis. The principal findings of all selected articles without food industry involvement were examined and classified in the same way as the principal findings of the articles with food industry involvement. As there was no specific industry actor involved in these articles, a broad interpretation of food industry interests was taken when assessing the extent to which articles favoured food industry interests. For example, a favourable finding for any food product or nutrient was considered favourable to the food industry, whereas a negative finding for any food product or nutrient was considered unfavourable. The primary topic area of each of the articles was noted, including the particular foods, food components or nutrients (as relevant).

Assessments of principal findings were conducted independently by two of the authors (DR and SD), with any discrepancies discussed and resolved with a third author (GS). Results were analysed by type of food industry involvement and by journal. For the purposes of this analysis of 'type of food industry involvement', author affiliations with the food industry and direct funding for the study from the food industry were grouped together (as they were considered more direct involvement) and compared to other types of food industry funding (that were considered less direct involvement).

## Statistical analysis

All articles with food industry involvement were identified from each of the ten included journals, with the frequency and percentage in each category of favourability calculated. For the randomly selected matched sample of research articles with no food industry involvement, we calculated 95% confidence intervals (CIs) for the proportion of articles in each category of favourability (e.g., favourable or unfavourable to food industry interests) using Stata 15.0 (StataCorp, College Station, TX, USA).

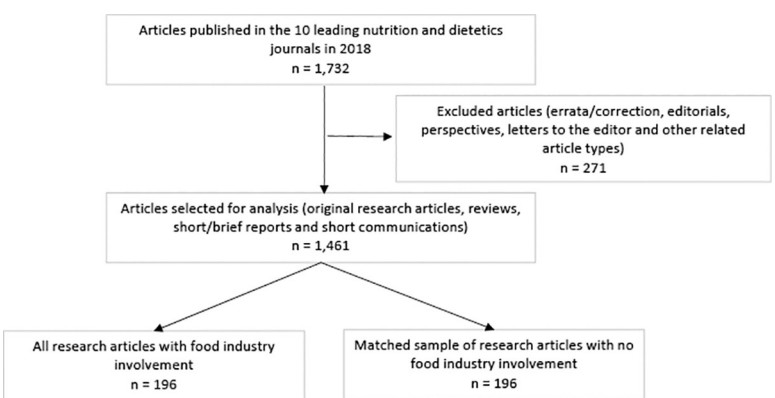

**Fig 1. Study flow diagram indicating the number of articles included in the study.**

## Results

Of the 1,732 articles published in the selected journals, 1,461 peer-reviewed research articles met our inclusion criteria (n = 271 excluded) (**Fig 1**). Amongst these, 196/1,461 (13.4%) were classified as having food industry involvement (**Table 2**). Refer to **S2 Table** for details of food industry actors identified.

**Table 2. Original research articles with food industry involvement in the top 10 nutrition-related journals in 2018.**

| Journal | Articles included in the sample | Articles with authors reporting affiliations related to the food industry [1] (n, % of row total) | Articles with declared funding from the food industry | | | Articles with other [2] stated involvement from the food industry (n, % of row total) | Total number of articles with food industry involvement (n, % of row total) [3] |
|---|---|---|---|---|---|---|---|
| | | | Direct funding for the study (n, % of row total) | Donation of products to be used for the study (n, % of row total) | Funding received for other research not directly related to the study (n, % of row total) | | |
| *The Journal of Nutrition* | 223 | 22, 9.9% | 46, 20.6% | 8, 3.6% | 12, 5.4% | 35, 15.7% | 63, 28.3% |
| *Nutrition Reviews* | 53 | 5, 9.4% | 7, 13.2% | 0, 0.0% | 6, 11.3% | 12, 22.6% | 13, 24.5% |
| *The American Journal of Clinical Nutrition* | 221 | 11, 5.0% | 25, 11.3% | 9, 4.1% | 20, 9.0% | 8, 3.6% | 37, 16.7% |
| *Clinical Nutrition* | 212 | 9, 4.2% | 20, 9.4% | 10, 4.7% | 16, 7.5% | 11, 5.2% | 35, 16.5% |
| *Obesity* | 231 | 3, 1.3% | 8, 3.5% | 5, 2.2% | 8, 3.5% | 8, 3.5% | 22, 9.5% |
| *Advances in Nutrition* | 64 | 3, 4.7% | 4, 6.3% | 1, 1.6% | 2, 3.1% | 2, 3.1% | 6, 9.4% |
| *Nutrition Research Reviews* | 21 | 1, 5.8% | 0, 0.0% | 0, 0.0% | 0, 0.0% | 1, 4.8% | 1, 4.8% |
| *International Journal of Obesity* | 206 | 3, 1.5% | 3, 1.5% | 0, 0.0% | 4, 1.9% | 3, 1.5% | 10, 4.9% |
| *International Journal of Behavioural Nutrition and Physical Activity* | 124 | 1, 0.8% | 4, 3.2% | 0, 0.0% | 1, 0.8% | 1, 0.8% | 5, 4.0% |
| *Paediatric Obesity* | 106 | 1, 0.9% | 3, 2.8% | 0, 0.0% | 1, 0.9% | 1, 0.9% | 4, 3.8% |
| **Total** | **1461** | **59, 4.0%** | **120, 8.2%** | **33, 2.3%** | **70, 4.8%** | **82, 5.6%** | **196, 13.4%** |

[1] Food industry includes: (1) all private sector organisations involved in food and beverage production, distribution, marketing and retail; (2) manufacturers of nutrition supplements and breast-milk substitutes; (3) relevant industry groups and trade associations; (4) organisations that receive the majority of their funding from organisations in the food industry.

[2] Articles with other involvement include involvement noted in acknowledgments and those that did not fit within the above categories.

[3] Total does not represent the sum of the previous columns due to instances where food industry involvement occurred in a number of categories.

The most common form of involvement was the provision of direct funding for the study (n = 120/196, 61.2%). Other involvement (including acknowledgments and information listed in the conflict of interests section and not related to other categories) represented the second most common form of involvement (82/196, 41.8%) followed by industry funding received for other research not directly related to the study (70/196, 35.7%) and authorship (59/196, 30.1%) (**Table 2**).

Food industry involvement was noted across all 10 journals included in the sample. *The Journal of Nutrition* (28.3%), *Nutrition Reviews* (24.5%), and *The American Journal of Clinical Nutrition* (16.7%) published the highest proportion of articles with food industry-involvement. *Paediatric Obesity* (3.8%), *International Journal of Behavioural Nutrition and Physical Activity* (4.0%), and *International Journal of Obesity* (4.9%) published the lowest proportion of articles with food industry involvement (**Table 2**). Each journal had similar policies in place that required authors to disclose conflicts of interest. Four journals (*Advances in Nutrition*, *The Journal of Nutrition*, *Obesity*, *Paediatric Obesity*) included statements regarding conflicts of interest of their editorial board on the journal website. Editors from six journals (*The American Journal of Clinical Nutrition*, *Advances in Nutrition*, *International Journal of Obesity*, *Nutrition Reviews*, *The Journal of Nutrition*, *Obesity*) were identified as having involvement with the food industry (see **S3 Table**). No other relevant policies regarding studies with food industry involvement were identified by any journal.

A diverse range of sectors of the food industry were involved in the research assessed (**Table 3**). The sectors most often represented were processed food manufacturing (39.3%), dietary supplement manufacturing (28.6%) and dairy (27.0%). Food retailers (including supermarkets) were involved in the fewest papers (2.6%). Of the 161 food industry actors identified as involved in research articles, the highest proportion (41.6%) were classified as trade/industry associations, 35.4% were classified as small corporations/other entities, and 23.0% were classified as large corporations (**S4 Table**). However, these large corporations were the most frequently involved (47.8% of identified instances of food industry involvement), followed by trade/industry associations (36.4% of identified instances of food industry involvement) and small corporations/other entities (15.8% of identified instances of food industry involvement)

**Table 3. Food industry involvement in research articles in the top 10 nutrition-related journals in 2018, by food industry sector.**

| Food industry sector[1] | Number of articles specifying food industry involvement, % of total[2] |
|---|---|
| Processed food manufacturing | 77, 39.3% |
| Dietary supplement manufacturing | 56, 28.6% |
| Dairy | 53, 27.0% |
| Primary production (non-dairy, non-meat) | 43, 21.9% |
| Other | 30, 15.3% |
| Non-alcoholic beverage manufacturing | 23, 11.7% |
| Meat and livestock | 12, 6.1% |
| Food chemical suppliers and food technology companies | 6, 3.1% |
| Food retail | 5, 2.6% |

[1] See **S1 Table** for food industry sector definitions and **S2 Table** for a list of identified organisations within each food industry sector.

[2] In many cases, multiple food industry actors were involved in a single article. See **S5 Table** for further details of involvement of individual food industry actors.

**Table 4. Nature of the findings in articles with and without [1] food industry involvement, by type of food industry involvement.**

| | Articles with food industry involvement [2] | | | Articles with no food industry involvement [1] n, % of column total (95% Confidence Intervals) |
|---|---|---|---|---|
| | Articles with authors reporting affiliations related to the food industry OR direct funding for the study from the food industry n, % of column total | Articles with no reported author affiliations related to the food industry AND no direct finding for the study from the food industry n, % of column total | Total n, % of column total | |
| Articles with findings **favourable** to the food industry | 86, 66.2% | 23, 34.9% | 109, 55.6% | 19, 9.7% (7.0-12.4%) |
| Articles with findings **unfavourable** to the food industry | 6, 4.7% | 7, 10.7% | 13, 6.6% | 12, 6.1% (3.1%-10.6%) |
| Articles with **mixed** findings with respect to the food industry | 8, 6.2% | 11, 16.7% | 19, 9.7% | 20, 10.2% (6.8%-13.9%) |
| Articles with **neutral** findings with respect to the food industry | 2, 1.5% | 3, 4.6% | 5, 2.6% | 28, 14.3% (10.9%-17.8%) |
| Articles with findings **not applicable** to food industry interests | 28, 21.5% | 22, 33.3% | 50, 25.5% | 117, 59.7% (54.5%-65.7%) |
| **Total** | **130, 100%** | **66, 100%** | **196, 100%** | **196, 100%** |

[1] A random sample of articles without food industry involvement were selected to match the number of articles with food industry involvement for each journal included in the study.

[2] 95% confidence intervals are not provided for articles with food industry involvement because we identified all articles with declared food industry involvement from the population of articles in the selected journals.

(**S4 Table**). Refer to **S5 Table** for further information on the industry actors identified as being involved in more than 1% of articles.

The majority of papers with food industry involvement reported findings that were considered favourable to the food industry (n = 109, 55.6%) (**Table 4**). The proportion of articles with findings considered favourable to the food industry was even higher (66.2%) where study authors reported either affiliations related to the food industry or direct funding for the study from the food industry (**Table 4**). In contrast, of the 196 randomly selected articles with no identified food industry involvement, 19 (9.7%, 95% CI: 7.0–12.4) reported findings classified as favourable to the food industry. The vast majority (n = 15/19, 78.9%) of these articles related to particular nutrients and/or food components (e.g., protein, vitamins), with the remaining four articles (21.1%) relating to foods and food products (e.g., coffee, green tea) (**S6 Table**).

Only a small proportion (n = 13, 6.6%) of papers with food industry involvement reported results that were unfavourable to the food industry (**Table 4**). The percentage of articles with findings unfavourable to the food industry or mixed findings were similar for those articles with and without food industry involvement (**Table 4**). 117 (59.7%, 95% CI: 54.5–65.7) articles with no food industry involvement had findings considered not applicable to the food industry, compared to 50 (25.5%) of the articles with food industry involvement. Similar patterns were observed across each journal (**S7 Table**).

## Discussion

This study found that 13.4% of peer-reviewed research articles in the top 10 most-cited nutrition- and dietetics-related journals from 2018 reported food industry involvement. Food industry involvement spanned a number of industry sectors, with processed food

manufacturing, dietary supplement manufacturing and dairy most often represented. The vast majority of industry involvement was from large corporations and trade/industry associations, rather than smaller corporations. The proportion of articles with findings considered favourable to the food industry was substantially higher among those articles with food industry involvement (55.6%) compared to a random sample of those without (9.7%), with the difference even more marked where industry involvement in studies was more direct (author affiliations or direct funding for the study). The percentage of articles considered unfavourable to the interests of the food industry was similar among the articles with food industry involvement and the random sample of those articles without.

Considerable variation in the percentage of articles with industry involvement was observed between journals. *The Journal of Nutrition* and *Nutrition Reviews* published the highest proportion of articles with industry involvement. Both of these journals have declared connections to the food industry. Several members of the board of *The Journal of Nutrition* have declared conflicts of interest involving food companies [34]. *The Journal of Nutrition* is published by the American Society of Nutrition (ASN), which has formal partnerships with multiple food companies [35] and has been criticised for supporting food industry objectives over public health interests [24]. Other journals included in the sample (*The American Journal of Clinical Nutrition* and *Advances in Nutrition*) are also published by ASN, and had lower proportions of articles with food industry involvement compared to *The Journal of Nutrition*. *Nutrition Reviews* is published by the International Life Science Institute (ILSI), who were founded and are solely funded by large food industry companies including Mars, Nestlé, Coca-Cola and PepsiCo with the majority of their members' interests opposing public health policy and objectives [36, 37]. Future research should explore the extent to which a journal's connections to the food industry influence their publication priorities and editorial processes.

The findings in this study support existing evidence that research with food industry involvement is generally favourable to the interests of the food industry [8, 11, 15, 18, 21, 24, 26, 38, 39]. In particular, this study adds to the growing empirical evidence that food industry involvement in nutrition research likely influences research agendas to focus disproportionately on topics of importance to the industry, potentially at the expense of topics of greater public health importance [8, 18]. A recent scoping review by Fabbri and colleagues [18] demonstrated the impact of industry involvement across a range of diverse sectors (including medicine and nutrition), finding that industry-funded research was more often focused on products, processes or activities that can be commercialised and marketed, rather than non-market based activities. They concluded that "corporate interests can drive research agendas away from questions that are the most relevant for public health" [18]. In addition, food industry-funded research has been noted as often focusing on a specific nutrient, potentially enabling the funder to market the benefits of particular nutrients [24]. While it has previously been reported that nutrition research funded by the food industry typically respects scientific standards for conducting and reporting scientific studies [17], the food industry was itself involved in that assessment, and the issue warrants further detailed exploration.

It has been well documented that a range of industries, including the food industry, seek involvement in research, develop research that is favourable to their interests, and make use of scientific evidence as part of broader efforts to influence public health policy [19, 22, 29, 40–42]. Moreover, there is evidence that major corporations have pushed for policy making systems that provide a route for feeding corporate evidence into policy making [42, 43]. There are several examples of topic areas in which research funded by the food industry favours particular products or diverts attention away from a public health issue. For example, with respect to sugar-sweetened beverages (SSBs), a body of research suggests that the involvement of the SSB sector in research has resulted in research that reports favourable findings for the industry [11,

44]. In addition, researchers have documented instances where Coca-Cola maintained control over study data and the disclosure of results for research it funded. Some research agreements between the company and their contracted researchers stated that Coca-Cola had the ultimate choice regarding publication of research findings [45].

## Study limitations

To date, this is the first study to systematically examine the extent of involvement of the food industry in peer-reviewed research articles published in the leading nutrition and dietetics journals. Importantly, much peer-reviewed nutrition research is published outside of the selected nutrition and dietetics journals. Moreover, the study was not designed to identify research with food industry involvement that is published in topic areas outside of nutrition and dietetics, outside of peer-reviewed journals, or that is funded or conducted by the industry but remains unpublished. Accordingly, the study represents only a small and selected analysis of the extent of food industry involvement in nutrition research. Future studies should investigate nutrition-related articles from journals with both a nutrition and non-nutrition focus (including, for example, journals in medicine and public health). Ways to automate methods for comprehensively identifying different types of food industry involvement in published studies need to be explored.

The classification of the principal findings of studies as favourable or unfavourable to the interests of the food industry was based on the knowledge of the researchers involved, which may have led to instances of unintended misclassification. Given the magnitude of the differences observed between articles with and without food industry involvement, unintended misclassifications are highly unlikely to have impacted the overall conclusions.

We did not perform any analysis by study design of the included articles or in relation to the appropriateness and rigour of the research methods used in each article. Accordingly, we did not assess the influence of food industry involvement on scientific methods or the way in which they were applied. Aspects of study design and specific mechanisms by which food industry involvement may influence study focus areas and results should be included in future studies.

The analysis relied primarily on the self-disclosure of food industry involvement (through declared conflicts of interests, funding acknowledgments, and author affiliations), with different journals having different disclosure requirements. We did not conduct an analysis of the veracity of each journal's conflict of interest disclosure requirements, but this warrants further exploration. Importantly, undisclosed food industry involvement cannot be captured using the approach we adopted in this study. There is evidence that the disclosure of conflict of interest is under-reported in research [45, 46], indicating that the percentage of articles with food industry involvement may be larger than that observed here. In addition, our identification of food industry organisations involved in the included studies may have been incomplete. While we made use of an established list of food industry front groups as well as online searches of identified organisations to determine the nature of their operations and funding sources, it has previously been noted that financial links to the food industry are often not publicly available [30].

Finally, we did not conduct a detailed examination of the extent to which the editors of each journal have links to the food industry. Future research should further explore links between journal editors and the food industry and the role of journal editors in assessing conflicts of interest with the food industry.

## Implications of the findings

The finding that food industry involvement is commonplace in peer-reviewed research in leading nutrition-related journals has several implications. With increased recognition of food

industry bias within research, it is important to consider ways of maximising the integrity of research published in respected peer-reviewed nutrition journals and ensuring that research focused on issues of public health relevance is prioritized. One option could be to limit industry funding of research to a government- or independently-controlled pool of money that supports a research agenda developed independent of industry, with strict processes to ensure freedom from industry influence [47]. A similar model for pharmaceutical research already operates in Italy [48], and in relation to the tobacco and alcohol industry in California and Thailand [49].

Further, it is important that research institutions have strict, regularly updated and transparent guidelines and policies to regulate and report on their engagement with industry, including specifying the level of engagement permitted with different actors. For those institutions with food industry involvement, processes need to be put in place to ensure that the potential influence of the food industry on research agendas and research methods are managed [50]. Example of guidelines for managing engagement with industry include those from the Charles Perkins Centre at the University of Sydney [51] and the Global Obesity Centre at Deakin University in Australia [52].

Journals could also consider adopting detailed policies regarding articles with declared food industry involvement. Such policies could place limits on the number of articles that the journal will accept for review, specific topic areas where food industry involvement is discouraged, or specific sections in journals for studies with industry involvement [24]. Based on the findings of this study, all articles that include any type of food industry involvement warrant close scrutiny from journals, with a particular focus on more direct types of involvement (e.g., author affiliations and direct funding for a study). Journals should also have clear policies on disclosing editorial conflicts of interest, including any links between editors and the food industry. Moreover, any such conflicts need to be actively managed or eliminated. Further, research that investigates appropriate standards of disclosure and involvement can guide policy and practice in this area.

## Conclusion

Food industry involvement in peer-reviewed nutrition research is commonplace, and the results of the majority of studies with food industry involvement favour the interests of the food industry. Given the potential competing interests of the food industry on the one hand, and scientific and population health interests on the other, it is important to explore mechanisms that can safeguard the integrity and public relevance of nutrition research, and ensure they are not undermined by the influence of the food industry.

## Supporting information

**S1 Table. Definitions of categories used to classify organisations from the food industry.**
(DOCX)

**S2 Table. Food industry actors identified as involved in research studies in the top 10 most-cited nutrition- and dietetics-related journals in 2018, by food industry sector and actor classification.**
(DOCX)

**S3 Table. The top 10 most-cited nutrition- and dietetics-related journals in 2018 and their declared involvement with the food industry.**
(DOCX)

**S4 Table. Food industry actors identified as being involved in the top 10 most-cited nutrition- and dietetics-related journals in 2018.**
(DOCX)

**S5 Table. Food industry actors identified as being involved in more than 1% of articles examined in the top 10 most-cited nutrition- and dietetics-related journals in 2018.**
(DOCX)

**S6 Table. Primary topic area of the random sample of articles without food industry involvement[1].**
(DOCX)

**S7 Table. Nature of the findings in articles with and without[1] food industry involvement, by journal.**
(DOCX)

## Acknowledgments

The authors would like to acknowledge the contribution of Benjamin Sullivan, an Honours student in the School of Health and Social Development at Deakin University in 2015, whose research informed the design of this study.

## Author Contributions

**Conceptualization:** Gary Sacks, Melissa Mialon, Adrian J. Cameron.

**Formal analysis:** Gary Sacks, Devorah Riesenberg, Melissa Mialon, Sarah Dean.

**Investigation:** Gary Sacks, Devorah Riesenberg, Adrian J. Cameron.

**Methodology:** Gary Sacks.

**Project administration:** Gary Sacks.

**Writing – original draft:** Gary Sacks.

**Writing – review & editing:** Gary Sacks, Devorah Riesenberg, Melissa Mialon, Sarah Dean, Adrian J. Cameron.

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
