## [Decision Letter · Decision Letter 0]

23 Jul 2020

PONE-D-20-18349

The nature and extent of food industry involvement in nutrition research: an analysis of peer-reviewed research articles from leading nutrition-related journals

PLOS ONE

Dear Dr. Sacks,

Thank you for submitting your manuscript to PLOS ONE. After careful consideration, we feel that it has merit but does not fully meet PLOS ONE’s publication criteria as it currently stands. Therefore, we invite you to submit a revised version of the manuscript that addresses the points raised during the review process.

The reviewers felt that this paper makes an important contribution. However, there are a few major issues that require attention:

First, the analysis feels incomplete. In providing a survey of the extent/prevalence of industry involvement in nutrition research, I would expect to see calculations of population proportions with confidence intervals, for example. In measuring the relationship between industry involvement and favourable results, the authors should perform statistical tests of these correlations. Sensitivity testing for non-disclosure, and also different types of industry involvement would strengthen the analysis.

In the Introduction, the paper would be strengthened by adding discussion of the complexity of these issues. For example, unlike Big Tobacco, the 'food industry' encompasses a wide range of entities; it would be helpful to provide a lay of the land and point out that many of the negative public health impacts you describe pertain to multinational corporations. I would also suggest considering how the size/scope of entity (corporation vs trade association vs small company) might be incorporated into your coding. Similarly, greater specificity on the public health impacts and the nature of "food industry interests" would make the impetus for and impact of the study clearer.

At points, the authors overstate the contribution this kind of analysis can make. Please clarify that the gap your study fills is the extent, nature and impact of food industry involvement in "published" or "peer reviewed" nutrition research. Presumably, a great deal of industrial research is conducted by the food industry.

Clarification is requested by the reviewers for a number of the coding categories.

I would suggest conducting a sensitivity analysis where you separate out industry funding/employment from financial conflicts of interest with industry. The current analysis seems to elide over some important differences in mechanisms for industry influence. I would expect the latter category to also be more common and yet, perhaps the least influential, thus overstating the findings.

We look forward to receiving your revised manuscript.

Kind regards,

Quinn Grundy, PhD, RN

Academic Editor

PLOS ONE

Journal Requirements:

'I have read the journal's policy and the authors of this manuscript have the following competing interests: GS and AJC are academic partners on a publicly funded healthy supermarket intervention trial that includes Australian local government and supermarket retail (IGA) collaborators. GS has been involved in studies to benchmark the policies and commitments of food companies related to obesity prevention and nutrition in Australia, New Zealand, Canada, Malaysia and Europe. The authors have not received funding from any organization in the food industry. The authors have no other potential competing interests to declare.'

a. Please confirm that this does not alter your adherence to all PLOS ONE policies on sharing data and materials, by including the following statement: "This does not alter our adherence to  PLOS ONE policies on sharing data and materials.” (as detailed online in our guide for authors http://journals.plos.org/plosone/s/competing-interests).  If there are restrictions on sharing of data and/or materials, please state these.

Please note that we cannot proceed with consideration of your article until this information has been declared.

Reviewers' comments:

Reviewer's Responses to Questions

**Comments to the Author**

1. Is the manuscript technically sound, and do the data support the conclusions?

Reviewer #1: Yes

Reviewer #2: Partly

2. Has the statistical analysis been performed appropriately and rigorously? 

Reviewer #1: N/A

Reviewer #2: N/A

3. Have the authors made all data underlying the findings in their manuscript fully available?

Reviewer #1: Yes

Reviewer #2: Yes

4. Is the manuscript presented in an intelligible fashion and written in standard English?

Reviewer #1: Yes

Reviewer #2: Yes

5. Review Comments to the Author

Reviewer #1: OVERVIEW

Overall this is a useful and potentially important paper and should be published. I think it could be strengthened in a few ways. First, I think it is vital that clear definitions are given of “industry affiliated” and whether and how that was investigated. Second, I think the authors might be able to add a few further analyses that would strengthen the paper. I should note, however, that I don’t think these are essential. Finally, I think the tables could be improved to make the presentation simpler and clearer. I would also suggest greater attention be given to the variation in levels of favourable industry research by journal and role of journal editors – I see that as a key finding.

Detailed comments follow by section.

COMMENTS

Abstract

1. Method needs some more specifics eg does not say how identified articles were assessed for inclusion and coded.

2. Result – I would consider including % by journal because I think that is one of the most interesting and important findings – ie journals with FI links publish more FI research favourable to the FI

Methods

3. I think it would help to know each journal’s position on COI – are authors required to fully disclose COI? Do these journals have any policy on taking food industry-funded/linked research?

4. “food industry actors”: can you include the no. of article assessed (p 5)

5. Does the definition of food industry used have a source you can cite?– is it an accepted definition?

6. You then include “organisations affiliated with food industry organisations” - I think some clarity around definition here is needed: 1st a definition for “food industry organisations” is needed (this may simply be “ food industry” as you have just defined? Otherwise you need a new definition of “FI organisations”.

7. 2nd, I think you also need to define “affiliated” and clarify how you determine which additional organisations are ‘afffiliated’. Ie do you mean “funded by”? And did you need to investigate this? In my experience the funding to some of these organisations is hidden and there is a need to investigate financial links. Some tobacco industry research has developed systems for identifying these financial links which you could perhaps draw on.

8. Food industry involvement: You detail who did much of the coding – eg “two of the authors (DR and GS) to 112 determine whether there was food industry involvement in the paper” and “The categorisation was performed independently by two of the authors 128 (DR and SD), with any discrepancies discussed and resolved with a third author (GS).” But not who coded/double coded whether the findings were favourable.

9. You define a lot of coding categories in the text. It might be worth considering whether this would all be clearer in a table?

10. “The same number of articles with and without industry involvement in each journal was selected for analysis.” – was this overall or by journal? And, I assume this selection effectively came from the pile you had rejected in the 1st run through? Pls can you detail this and give numbers (fine if these go in results). Eg if 1450 articles published and assessed, 196 were food industry linked. The rest then you could select from as non-industry linked?

11. I’m wondering if there are a few minor things you could do to strengthen the paper:

a. In my experience authors might identify themselves as working for a corporation, or funded by a corporation but then declare no COI even when the paper focuses on a topic that is directly in that corporations area of interest (ie where there is in fact a COI). Did you record whether declarations of COIs were consistent with your assessment of whether there was a COI? If not, could you add this?

b. I’m interested in the varying percentages of industry linked papers by journal and am glad you present that. Just wondering if there is any evidence of certain editors having links to the FI and whether you would consider exploring and adding that? - Some of the percentages are very high. (I wrote this and then got to the discussion where you mention this! I wonder if it would be more powerful to more formally assess this and include it in the methods and results. Eg you coudl have a table with journals, % FI linked papers, the % of editors with Fi links, other journal links to FI etc.)

Results

12. Can you clarify if 1450 is the total no. of articles once editorials, letters excluded?

13. Linked to my query about on transparency, you seem to have taken all statements of funding and COI at face value – is that correct? Or did you investigate whether, for example, unknown funders were linked to industry?

14. Table 1 – can you give %ages in the bottom (totals) line

15. Table 3 – ditto – pls give the N total and not just % in the bottom (totals) line

16. “Of the 196 randomly selected articles with no industry declared food industry involvement,

17. 207 138 (70.4%) had findings considered not applicable to the food industry, 24 (12.2%) reported findings classified as favourable to food industry interests, and 13 (6.6%) reported findings categorised as unfavourable to food industry interests (Table 3).” – I think it is hard to see where these data are in table 3. Pls consider changing the structure of and/or headings in table 3 to make this clearer

Discussion

18. “It has been well documented that a range of industries, including the food industry, seek involvement in research and make use of scientific evidence as part of broader efforts to influence public health policy [16, 26-28].” – I think there are some other important papers to include there. In particular Ulucanlar’s Policy Dystopia Model which identifies “information management” as a key industry tactic – this includes producing & disseminating favourable research. You could also flag that diverse major corporations (including FI) have pushed for policy making systems that provide a route for feeding corporate evidence into policy making https://journals.plos.org/plosmedicine/article/comments?id=10.1371/journal.pmed.1000202 and indeed have gone on to use those systems (eg https://journals.plos.org/plosmedicine/article/comments?id=10.1371/journal.pmed.1000202). Ie the development of favourable evidence is part of a broader system of policy influence.

19. In terms of policy suggestions there is work by Joanna Cohen which explores models through with corporate (in this case tobacco) money can be used to fund research while avoiding COI. https://tobaccocontrol.bmj.com/content/18/3/228?ijkey=141181f5dd20d7990a3e6b5dbefa6957116251eb&keytype2=tf_ipsecsha. There are examples of these systems in California and Thailand (mentioned here https://www.bmj.com/content/353/bmj.i2161)

20. In terms of journal recommendations – if you explored whether COIs are actually declared when they exist you could make a recommendation of making clear COI statements. Surely you also need to recommend editors should not have FI links?

Reviewer #2: Bias in nutrition science is an important topic and an under-explored area of research. This article makes a contribution by examining food industry sponsorship/ author conflicts of interest in the top 10 nutrition/dietetics journals.

My first reaction to the abstract and introduction is that the authors have overstated the scope of the manuscript. For example, the authors state that "This study aimed to identify the extent of food industry involvement in peer-reviewed nutrition articles." Food industry involvement seems to be operationalized as the relationship between article sponsorship and author conflict of interest compared to favorable/unfavorable "principle findings." This manuscript does not evaluate methodological quality, nor does it evaluate the relationship between sponsorship/COI and study results --therefore it is not analyzing the full extent of food industry involvement. (Furthermore, it may not be possible to know the full extent of industry involvement without access to internal documents). This manuscript would be improved by clearly stating what is analyzed and how it contributes to the overall literature.

Introduction:

Line 63 - -It is not clear that the authors are aware of food industry involvement in influencing guidelines for evaluating integrity in nutrition science. For example, citation #8 here is partly sponsored by ASN and ILSI.

Paragraph at Line 67 -- the list of objectionable motives don't seem very objectionable. Why shouldn't a business entity conduct marketing research? What's wrong with sponsoring research that benefits industry interests?

Line 82-86: This is a superficial review of the literature and would benefit from more explanation and detail--to set the context for the current study and explain how it fits in with current literature.

Line 86: "To date, no study has comprehensively examined the extent, nature and impact of food industry involvement in nutrition research." --Again, this is an overstatement. No one study could possibly do this.

Line 88: "This study aimed to fill this significant evidence gap by systematically identifying the extent of food industry involvement in peer-reviewed research articles from a large sample of leading nutrition-related journals." Again -- no one study could fill this gap. This manuscript contributes to a growing body of empirical analysis. It is also important to acknowledge that much "nutrition research" takes place outside of nutrition/dietetics journals.

Sample:

The current SJR list has Annual Review of Nutrition as #1 and Pediatric Obesity at #14 – recommend clarifying at what point in time list was accessed.

Line 144: the use of "principal findings" is a bit confusing and does not conform with past evaluations (to my knowledge). If the authors are using an established methodology -- should be referenced. Otherwise -- it is unclear to talk about results contained within conclusions. Results are results. Conclusions are conclusions. See Citation #5

Results:

Line 183: Results would be improved with inclusion of a study flow diagram. Did the authors really assess 1450 articles, or were they screened for food industry involvement and article type? A flow diagram would make it easier to understand the random sample of non-industry involved articles which right now reads like an add on requested by a previous reviewer. The results would also benefit from aligning the flow of tables to match flow of text. Results/tables would also be easier to follow if both n/% are included in Total rows.

Discussion:

Line 214 -- would be more accurate to specify that articles without food industry involvement = a random sample.

Line 226--"importantly, however" -- it is not clear what distinction is being made here.

Line 236 -- "It has previously been reported that nutrition research funded by the food industry typically respects scientific standards for conducting and reporting scientific studies [14]" This is an industry sponsored study. It should not be taken at face value.

Line 276 - is this a limitation or a continuation of the discussion?

6. PLOS authors have the option to publish the peer review history of their article (what does this mean?). If published, this will include your full peer review and any attached files.

Reviewer #1: No

Reviewer #2: No

---

## [Author Response · Author response to Decision Letter 0]

31 Aug 2020

Please refer to point by point response and detailed comments in 'response to reviewers' file attached.

---

## [Decision Letter · Decision Letter 1]

9 Oct 2020

PONE-D-20-18349R1

The nature and extent of food industry involvement in nutrition research: an analysis of peer-reviewed research articles from leading nutrition-related journals

PLOS ONE

Dear Dr. Sacks,

Thank you for submitting your manuscript to PLOS ONE. After careful consideration, we feel that it has merit but does not fully meet PLOS ONE’s publication criteria as it currently stands. Therefore, we invite you to submit a revised version of the manuscript that addresses the points raised during the review process.

While one reviewer confirmed that their previous comments had been addressed, unfortunately, not all of the original reviewers were available to re-review the manuscript. Thus, I invited additional reviewers to review the re-submission and ask that you attend to their comments. Particularly, I ask that you re-consider and/or justify the approach to classify a "favourable" conclusion among non-industry-sponsored studies. Second, two reviewers have raised concerns about the adequacy of the sampling methods and their reporting and whether these constitute a 'systematic' assessment of the literature; at minimum, please address this in the study's limitations, or, consider revising these methods.

The third reviewer raised the question of conducting a regression analysis to infer a causal relationship, which I suggest is beyond the scope of this design, however, I would appreciate your response.

We look forward to receiving your revised manuscript.

Kind regards,

Quinn Grundy, PhD, RN

Academic Editor

PLOS ONE

Reviewers' comments:

Reviewer's Responses to Questions

**Comments to the Author**

1. If the authors have adequately addressed your comments raised in a previous round of review and you feel that this manuscript is now acceptable for publication, you may indicate that here to bypass the “Comments to the Author” section, enter your conflict of interest statement in the “Confidential to Editor” section, and submit your "Accept" recommendation.

Reviewer #2: All comments have been addressed

Reviewer #3: (No Response)

Reviewer #4: (No Response)

2. Is the manuscript technically sound, and do the data support the conclusions?

Reviewer #2: Yes

Reviewer #3: Yes

Reviewer #4: Partly

3. Has the statistical analysis been performed appropriately and rigorously? 

Reviewer #2: Yes

Reviewer #3: Yes

Reviewer #4: Yes

4. Have the authors made all data underlying the findings in their manuscript fully available?

Reviewer #2: Yes

Reviewer #3: Yes

Reviewer #4: Yes

5. Is the manuscript presented in an intelligible fashion and written in standard English?

Reviewer #2: Yes

Reviewer #3: Yes

Reviewer #4: Yes

6. Review Comments to the Author

Reviewer #2: (No Response)

Reviewer #3: In an important contribution to understanding of the role of commercial actors in science, the authors map the extent and nature of food industry involvement in the area of nutrition research. They identify peer-reviewed articles which declare food industry funding or links, in 10 leading nutrition journals, and categorise whether their findings are favourable the food industry. They compare the latter to a sample of articles which do not declare industry involvement, concluding that articles with industry involvement are more likely to be favourable to the food industry than those without. They also find that the extent of food industry involvement nutrition-related research is substantial, though it varies between journals.

Overall, the paper is well written, a pleasure to read, and fills a crucial research gap. The rationale is clear and, for the most part, the authors explain very well what they did. The classification of food industry involvement is very clear and detailed, which provides interesting insights into which parts of the industry are most involved in nutrition research and will be useful for other researchers looking to classify food industry actors.

The key point I would like to raise pertains to the analysis of the articles without food industry involvement for favourability. The definition used for the industry-funded/linked articles seems appropriate as the favourability of their findings can be assessed in relation to the associated food industry actor. Favourability for the articles without industry involvement, however, is defined as follows: “a favourable finding for any food product or nutrient was considered favourable to the food industry, whereas a negative finding for any food product or nutrient was considered unfavourable” (line 188 onwards). This approach to operationalising favourability to the food industry is very broad and raises the question whether articles supporting the consumption of nutrients and products widely considered as healthy, such as whole grains, fruits, and vegetables, would be included as favourable to the food industry. I suggest that the authors refine this part of the paper accordingly. This could be done, for example, by clarifying or revisiting the definition of favourability for articles without food industry involvement, or by expanding the reporting of results on favourability for articles without industry involvement (i.e. which products/nutrients did the findings support).

Additionally, it would be helpful to have more detail regarding the process behind the selection of the random sample of articles without food industry involvement (line 183).

Reviewer #4: Sacks and colleagues have conducted an important piece of work to describe characteristics, such as authors' affiliations, funding source from food industry, and research findings regarding the food intervention/exposure, in academic articles published in the top 10 nutrition journals. While it is important to describe these characteristics, it is also important to understand whether there is any causal inference can be established in this setting.

Specific comments in the text of the manuscript:

Abstract

The study has used nature and extent in the objective of the study frequently. However, this is not clear to the readers what the authors are trying to measure. Suggest using more precise wording to describe what the study measured.

Methods: if a systematic search was applied to identify the articles, please state it here. Otherwise, please describe how the articles were selected and what are the principal search articles and a random sample of studies.

Results: some of the values should be provided. For example, please provide values for the variation related to “196/1,461 (13.4%) articles reported food industry involvement, with large variation by journal.” and values of proportion for “Journals with declared links to the food industry published a higher proportion of papers with food industry involvement.”

Conclusion: it needs more substantive evidence, such as regression analysis, to substantiate the conclusion: “This study, in line with previous literature, has shown that the results of peer-reviewed studies involving the food industry are more likely to favour relevant food industry interests than peer-reviewed studies without food industry involvement.”

Main text:

Introduction:

line 93 on page 11: Again, I find it hard to understand what it means or what is measured here by using "extent" and "nature" in the objective of the study.

Line 95: “Better information in this area” – please be specific about what “better” information means here. This will also help the readers understand the research gap in the literature.

Methods:

Please provide rationale of only looking into 10 nutrition journals rather than conducting a systematic review to identify nutrition studies. I would also recommend describing the study design and intervention/expsoure of the included studies.

Details are needed to describe how the information was obtained in line 119-123. Since the authors are assessing how industry involvement in nutrition research may affect “favorable” findings, is it possible to conduct a regression analysis to infer a potentially causal relationship?

Results

Table 2: can the authors provide 95% CI for the articles with food industry involvement, so that it is comparable with the results in Table 4?

Table 3: what does “other” include?

Discussion

I think one of the major limitations of the study is the selection of articles. The current inlcuded studies may not be complete, as many nutrition-related research articles are not necessarily published in nutrition journals but in other journals with a broader category in health. These may include high-impact medical journals and science journals.

7. PLOS authors have the option to publish the peer review history of their article (what does this mean?). If published, this will include your full peer review and any attached files.

Reviewer #2: No

Reviewer #3: No

Reviewer #4: No

---

## [Author Response · Author response to Decision Letter 1]

27 Oct 2020

Please refer to separate document (attached) that has a point by point response to each comment

---

## [Decision Letter · Decision Letter 2]

17 Nov 2020

The characteristics and extent of food industry involvement in peer-reviewed research articles from 10 leading nutrition-related journals in 2018

PONE-D-20-18349R2

Dear Dr. Sacks,

We’re pleased to inform you that your manuscript has been judged scientifically suitable for publication and will be formally accepted for publication once it meets all outstanding technical requirements.

Kind regards,

Quinn Grundy, PhD, RN

Academic Editor

PLOS ONE

Additional Editor Comments (optional):

Reviewers' comments:

Reviewer's Responses to Questions

**Comments to the Author**

1. If the authors have adequately addressed your comments raised in a previous round of review and you feel that this manuscript is now acceptable for publication, you may indicate that here to bypass the “Comments to the Author” section, enter your conflict of interest statement in the “Confidential to Editor” section, and submit your "Accept" recommendation.

Reviewer #3: All comments have been addressed

Reviewer #4: All comments have been addressed

2. Is the manuscript technically sound, and do the data support the conclusions?

Reviewer #3: Yes

Reviewer #4: Yes

3. Has the statistical analysis been performed appropriately and rigorously? 

Reviewer #3: Yes

Reviewer #4: N/A

4. Have the authors made all data underlying the findings in their manuscript fully available?

Reviewer #3: Yes

Reviewer #4: Yes

5. Is the manuscript presented in an intelligible fashion and written in standard English?

Reviewer #3: Yes

Reviewer #4: Yes

6. Review Comments to the Author

Reviewer #3: (No Response)

Reviewer #4: Thank you for putting the efforts to address my comments, which I am satisfied. Perhaps in future studies, consider a statistical regression model to see whether COI and conclusions in favor of sponsors would hold in these studies. This will definitely strengthen the argument and support the hypothesis.

7. PLOS authors have the option to publish the peer review history of their article (what does this mean?). If published, this will include your full peer review and any attached files.

Reviewer #3: No

Reviewer #4: No

---

## [Editor Report · Acceptance letter]

20 Nov 2020

PONE-D-20-18349R2 

The characteristics and extent of food industry involvement in peer-reviewed research articles from 10 leading nutrition-related journals in 2018 

Dear Dr. Sacks:

I'm pleased to inform you that your manuscript has been deemed suitable for publication in PLOS ONE. Congratulations! Your manuscript is now with our production department. 

Kind regards, 

on behalf of

Dr. Quinn Grundy 

Academic Editor

PLOS ONE